# Virus-Subtype-Specific Cellular and Humoral Immune Response to a COVID-19 mRNA Vaccine in Chronic Kidney Disease Patients and Renal Transplant Recipients

**DOI:** 10.3390/microorganisms11071756

**Published:** 2023-07-05

**Authors:** Astrid I. Knell, Anna K. Böhm, Michael Jäger, Julia Kerschbaum, Sabine Engl, Michael Rudnicki, Lukas Buchwinkler, Rosa Bellmann-Weiler, Wilfried Posch, Günter Weiss

**Affiliations:** 1Department of Internal Medicine II, Infectious Diseases, Immunology, Rheumatology, Pneumology, Medical University of Innsbruck, Anichstraße 35, 6020 Innsbruck, Austria; 2Institute of Hygiene and Medical Microbiology, Medical University of Innsbruck, Schöpfstraße 41, 6020 Innsbruck, Austria; 3Department of Internal Medicine IV (Nephrology and Hypertension), Medical University Innsbruck, Anichstraße 35, 6020 Innsbruck, Austria

**Keywords:** COVID-19, SARS-CoV-2, vaccination, vasculitis, chronic kidney disease, kidney transplant recipients

## Abstract

Patients with chronic kidney disease (CKD) or immunosuppression are at increased risk of severe SARS-CoV-2 infection. The vaccination of CKD patients has resulted in lower antibody concentrations and possibly reduced protection. However, little information is available on how T-cell-mediated immune response is affected in those patients and how vaccine-induced immune responses can neutralise different SARS-CoV-2 variants. Herein, we studied virus-specific humoral and cellular immune responses after two doses of mRNA-1273 (Moderna) vaccine in 42 patients suffering from CKD, small vessel vasculitis (maintenance phase), or kidney transplant recipients (KT). Serum and PBMCs from baseline and at three months after vaccination were used to determine SARS-CoV-2 S1-specific antibodies, neutralisation titers against SARS-CoV-2 WT, B1.617.2 (delta), and BA.1 (omicron) variants as well as virus-specific T-cells via IFNγ ELISpot assays. We observed a significant increase in quantitative and neutralising antibody titers against SARS-CoV-2 and significantly increased T-cell responses to SARS-CoV-2 S1 antigen after vaccination only in the CKD patients. In patients with vasculitis, neither humoral nor cellular responses were detected. In KT recipients, antibodies and virus neutralisation against WT and delta, but not against omicron BA.1, was assured. Importantly, we found no specific SARS-CoV-2 T-cell response in vasculitis and KT subjects, although unspecific T-cell activation was evident in most patients even before vaccination. While pre-dialysis CKD patients appear to mount an effective immune response for in vitro neutralisation of SARS-CoV-2, KT and vasculitis patients under immunosuppressive therapy were insufficiently protected from SARS-CoV-2 two months after the second dose of an mRNA vaccine.

## 1. Introduction

COVID-19, caused by severe acute respiratory syndrome coronavirus-2 (SARS-CoV-2), can lead to severe and life-threatening disease with respiratory insufficiency [1,2]. Specifically, patients with chronic diseases such as obesity, diabetes, cardiovascular, and renal disease, as well as under immunosuppression are at a higher risk for an adverse outcome [2,3,4,5,6]. Thus, it is particularly important to protect these patients at risk from a severe course of SARS-CoV-2 infection. Several vaccines have been developed to protect from SARS-CoV-2 infection, among which mRNA vaccines, i.e., mRNA-1273 (Spikevax, Moderna) or BNT162b2 (Corminaty, BionTech/Pfizer), have shown high efficacy against infection and a severe course of the disease [7,8]. However, the humoral immune response to vaccination was eventually low, particularly in patients at increased risk, including patients on hemodialysis [9], kidney transplantation (KT) recipients [9], and patients under immunosuppressive medication (e.g., glucocorticoids, rituximab) [10,11,12], or older patients [11]. Decisive components of an effective humoral immune response are not only the concentration of SARS-CoV-2 antibodies in the circulation but in particular the number of neutralising antibodies, and the neutralising capacity of those antibodies against actually circulating variants of SARS-CoV-2 [13,14]. However, for efficient immune defence against viral infections including SARS-CoV-2, cellular immune effector function orchestrated by T-cells is also crucial [15,16,17]. CD4+ helper T-cells mediate B-cell-induced antibody production and trigger anti-viral cellular immune responses, whereas CD8+ cytotoxic T-cells can target virus-infected cells and induce their apoptosis. Induction of SARS-CoV-2-specific CD4+ and CD8+ T-cells and higher initial IFNγ production by those cells have been shown to be associated with a milder course of COVID-19 [18]. In addition, the T-cell response to SARS-CoV-1 persisted longer after antigen contact than immune protection by antibodies and memory B-cells [19].

Moreover, an efficient T-cell immune function is important for protection from a severe course of SARS-CoV-2 (re-)infection [20]. Since the beginning of the pandemic, several virus mutations have emerged that differ in their potential to be neutralised by sera from convalescent or vaccinated people. In various studies, an immune escape from neutralising antibodies by new virus variants could be observed [17]. T-cell immunity may also be compromised, although it exerted a protective function in studies evaluating various virus subtypes in vaccinees due to its diversity of epitope recognition [17,21].

Therefore, the aim of this study was to investigate humoral and cellular immune responses and the neutralising capacity against different SARS-CoV-2 variants in patients with chronic kidney disease, vasculitis with renal involvement, and KT recipients after vaccination with two injections of an mRNA vaccine.

## 2. Materials and Methods

### 2.1. Study Design

This study initially involved 47 patients with chronic kidney disease (CKD), small vessel vasculitis with renal involvement, or previous kidney transplantation. Patients were vaccinated twice, receiving the first dose of Moderna mRNA-1273 in March/April 2021 and the second dose one month later. Written informed consent was obtained in accordance with the Declaration of Helsinki. Data collection and analysis were approved under ECS1236/2021 by the local ethics committee. The vaccination status was queried on the ELGA system (electronic health act), and patients’ characteristics (shown in Table 1) were obtained in the form of routinely collected clinical data. 

### 2.2. Blood Samples 

The first blood sample was taken a few minutes after the administration of the first dose of the mRNA-1273 vaccine (100 µg, day 1) and a second blood sample was drawn three months (median 91 days, min 90 days, max 91 days) after the first vaccination during routine check-ups at nephrology outpatient clinics. Blood was collected in EDTA and serum tubes; peripheral blood mononuclear cells (PBMCs) were obtained from EDTA blood using density gradient centrifugation with Pancoll human (25 min, 1250 rpm, room temperature). PBMCs were then washed with PBS (phosphate buffered saline) two times (5 min, 1250 rpm, 4 °C). The PBMCs were stored in a freezing mix suspension (45% FCS (fetal calf serum), 45% RPMI (Roswell Park Memorial Institute) medium, and 10% DMSO (dimethyl sulfoxide)) for <1 year at −80 °C. Quantitative determination of antibody levels (S1-RBD IgG in BAU/mL, Abbott Architect) was performed using chemiluminescence microparticle immunoassay (CMIA) at both time points. Neutralisation plaque assay and T-cell activation assay were carried out as previously described [18,21,22]. To evaluate the neutralising properties of serum antibodies, neutralisation tests against the virus variants SARS-CoV-2 wildtype (WT), delta (B1.617.2), and omicron (BA.1) were performed.

### 2.3. Neutralising Plaque Assay

VeroE6/TMPRSS2 cells (1.8 × 10^5^) were seeded in a 24-well plate with culture medium (DMEM high-glucose medium supplemented with 10% FCS, 1% L-Glutamine, and 1% Penicillin/Streptomycin; Sigma Aldrich, St. Louis, MO, USA). After incubation overnight at 37 °C, 5% CO_2_ serum samples were serial-diluted from 1:1 to 1:6250 and SARS-CoV-2 strains (1.5 × 10^4^ PFU/mL) were added (incubation for 1 h at 37 °C). After ultracentrifugation and resuspension in DMEM high-glucose medium with 5% FCS, 1% L-Glutamine, and 1% Penicillin/Streptomycin, VeroE6/TMPRSS2 cells were inoculated with antibody-opsonized SARS-CoV-2 (30 min on a shaker, RT, 30 min) and incubated (37 °C, 5% CO_2_). The inoculation medium was replaced afterwards with a culture medium (1.5% Low-melt Agarose, Biozym, Oldendorf, Germany) and the cells were incubated again (3 days, 37 °C, 5% CO_2_). Plaque visualization and counting were performed using 0.1% Neutral Red solution (3 h, Sigma Aldrich, MO, USA).

### 2.4. T-Cell Activation Assay

T-cell activation was examined by analysing INFγ production by PBMCs via pre-coated human SARS-CoV-2-specific IFNγ enzyme-linked immunospot (ELISpot) kits (AutoImmun Diagnostika). ELISpot MultiScreen^®^HTS 96-well filter plates (Millipore, Billerica, MA, USA) were activated with ethanol (35%), washed, and coated overnight with anti-human IFNγ monoclonal antibody 1-D1K (2 µg/mL; Mabtech, Stockholm, Sweden). We removed the coating solution before starting analyses and saturated the plates with D-PBS containing 10% FCS (Sigma Aldrich, MO, USA) for 2 h at room temperature. A total of 5 × 10^5^ PBMCs/well was counted and seeded in RPMI supplemented with 5% heat-inactivated human AB serum and 1% l-glutamine (Sigma Aldrich, MO, USA). Cells were stimulated using 0.6 nM/mL PepTiviator^®^ SARS-CoV-2 Peptide Pools (Miltenyi Biotec, Bergisch Gladbach, Germany) of spike glycoprotein (S/S1) in duplicates. As positive controls, a peptide pool (consisting of cytomegalovirus/Epstein–Barr virus/influenza virus; 2 µg/mL; Mabtech, Stockholm, Sweden) and phorbol myristate acetate (PMA)/ionomycin cell activation cocktail (1:500; Biolegend) were used for cell stimulation. PBMCs were seeded with a culture medium only to determine the background level for each donor. Following overnight incubation at 37 °C and 5% CO_2_, IFNγ production was detected using biotinylated anti-human IFNγ monoclonal antibody 7-B6-1 (1 µg/mL in Dulbecco’s PBS containing 0.5% FCS, Mabtech, Stockholm, Sweden) for 2 h at room temperature, followed by incubation of streptavidin–alkaline phosphatase (1:1000 in D-PBS containing 0.5% FCS; Sigma Aldrich, MO, USA), and finally treatment with 50 µL ready-to-use 5-bromo-4-chloro-3-indolyl phosphate/nitro blue tetrazolium (BCIP^®^/NBT) liquid substrate (Sigma Aldrich, MO, USA). After each step, the plates were washed five times with D-PBS. Positive spots were quantified using the ImmunoSpot analyser (Cellular Technology Limited, Shaker Heights, OH, USA) and spot quality was checked using ImmunoSpot software version 5.0.9.15.

### 2.5. Statistical Analysis

The statistical analysis was performed using LibreOffice Calc and R (Version 3.4.4). The Shapiro–Wilk test was used for testing normality. Non-parametric tests (Kruskal–Wallis test with Holm’s adjustment method, Wilcoxon test, and Wilcoxon test for independent groups/Mann–Whitney U test) were used with a significance level of *p* < 0.05. 

## 3. Results

Forty-seven participants with CKD, small vessel vasculitis (ANCA-positive with one exception), or KT recipients were included in the study with blood sampling and PBMC isolation on the day of vaccination. Of those, 43 patients received a second dose at an average interval of 35 days after the first dose. The four participants without a second vaccine dose were excluded from further analyses, as well as one patient with a positive in vitro nucleocapsid antibody result, indicative of infection prior to the second vaccination. This cohort then consisted of 18 CKD, 7 vasculitis, and 17 KT subjects and none of these patients had positive spike antibody detection prior to vaccination (Table 1). The median renal function impairment in CKD patients was stage G3 (G2: 1 patient; G3: 14 patients; G4: 3 patients). All vasculitis patients were at maintenance state. 

### 3.1. Immune Reaction to mRNA-1273—Quantitative Antibody Analysis

There were no measurable increases in antibodies in the three vasculitis (42.9%) and four KT (23.5%) subjects (spike antibodies < 3.0 BAU/mL) at three months after the first vaccination, respectively (see Appendix A). We found a significant increase in antibody titres within the disease groups (*p* < 0.05), except for the vasculitis group (*p* = 0.1003, Figure 1). The median antibody concentration after the second vaccination was 3572.9 BAU/mL in patients with CKD, 44.3 BAU/mL in those with vasculitis, and 27.8 BAU/mL in KT patients. Using the Kruskal–Wallis test, a significant difference in antibody levels after vaccination between the disease groups could be found (Table 1, *p* < 0.001). The post hoc analysis with Holm’s adjustment method showed a significant difference between CKD and KT (*p* = 1.4 × 10^−6^), and between CKD and vasculitis patients (*p* = 0.00039), which means that vasculitis and KT patients produced significantly fewer antibodies. A comparison of immune-suppressed versus non-immune-suppressed patients regardless of the underlying diagnosis is shown in Appendix A.

### 3.2. Neutralisation Assays

In the group of vasculitis patients in particular, only a small proportion showed neutralising antibodies (28.6% against the WT and delta variants), and there was hardly any neutralisation against omicron. By contrast, in CKD patients, neutralising antibodies were present against the WT and delta variants at 100%, respectively, and against omicron at a significantly lower level of 38.9%. In KT patients, no positive neutralising activity against the omicron variant was detected, whereas some patients were protected against the delta (23.5%) and WT (52.9%) variants according to the in vitro analysis. Figure 2 shows the percentage of patients in each group with positive, borderline, and negative results for the NT50 value three months after the first dose of vaccination. 

A 50% neutralisation titer (NT50) value ≥ 1:32 was interpreted as positive, 1:32 > NT50 ≥ 1:16 as borderline, and NT50 < 1:16 as a negative result [22]. The NT50 value before vaccination was negative for all variants in all patients with the exception of two KT patients for variant BA.1 (borderline), which was also recorded after vaccination (see Appendix A). The data were not normally distributed. Using the Kruskal–Wallis test, a difference in neutralisation capacity three months after vaccination can be determined between the cohorts (WT, delta, and omicron, *p* < 0.001). 

According to the post hoc analysis using the Wilcoxon test, significant differences in WT and delta variant neutralisation between CKD and vasculitis (*p* = 0.00035 and *p* = 0.0006), as well as KT patients (*p* = 2.3 × 10^−6^ and *p* = 1.9 × 10^−5^), were observed, which was also true for omicron neutralisation when comparing CKD and KT patients (*p* = 0.00058). Within the cohorts, there was no significant increase in neutralisation activity between the first vaccination (T1) and the three-month follow-up (T2) for the WT (*p* = 0.1814), delta (*p* = 0.3711), or omicron variants (*p* = 1) in vasculitis patients, whereas CKD patients showed a significant increase in neutralisation activity against all variants (WT and delta: *p* < 0.001; omicron: *p* = 0.0017). In KT patients, significant increases in viral neutralisation were found against the WT (*p* = 0.0059) and delta variants (*p* = 0.0143), but not against the omicron variant (*p* = 0.3711, Figure 3). 

In addition to the NT50 values, the 90% neutralisation titre (NT90) was determined, with an NT90 ≥ 1:16 interpreted as positive, 1:16 > NT90 ≥ 1:8 as borderline, and NT90 < 1:8 as negative. Appendix A shows the percentage of patients in each group with positive, borderline, and negative results for the NT90 value 3 months after the administration of the first vaccine dose. Appendix A shows the NT50 and NT90 data.

### 3.3. T-Cell-Mediated Immune Response 

We next evaluated the effects of vaccination on the induction of SARS-CoV-2-specific T-cell responses. For this, we performed an interferon-gamma (INFγ) enzyme-linked immunospot (ELISpot) assay to investigate SARS-CoV-2 spike glycoprotein subunit 1 (S1)-specific T-cell activation. The median of spot forming units and percentages of positive T-cell response on the day of the first vaccine dose and at follow-up are shown in Table 2, where a positive T-cell response was assumed for a value of ≥10 IFNγ spots/10^6^ PBMCs and a negative T-cell response for <9 IFNγ spots/10^6^ PBMCs. With the exception of vasculitis patients, we observed an increase in the number of patients with an S1-specific T-cell response. Notably, the majority of patients in the vasculitis group had a positive response to S1-specific T-cells even before vaccination (five of seven patients) which declined three months later (Table 2; for a detailed graphical presentation including *p*-values and significances, see Figure 4 below).

We also studied general T-cell activation upon stimulation of cells with a T-cell activation cocktail, as described in the Section 2. This indicated that T-cell activation, in general, was most prominent in vasculitis patients, although there was no significant difference between the disease groups at T1 (*p* = 0.5241; chi-squared = 1.292) and T2 (*p* = 0.1682; chi-squared = 3.5651). In addition, no significant difference in pan-T-cell activation was observed within the different disease groups from T1 to T2 (vasculitis: *p* = 0.297; CKD: *p* = 0.722; and KT: *p* = 0.222 Figure 4). 

By contrast, the specific T-cell response to S1 antigen stimulation increased significantly after vaccination in CKD patients only (*p* = 0.026, Figure 4). 

## 4. Discussion

This study investigated the immune response to mRNA-1273 vaccination in patients with vasculitis, CKD, and KT recipients on a humoral and cellular level from a longitudinal perspective. The data showed that the quantitative IgG antibodies against S1-RBD increased significantly after vaccination in CKD and KT patients, but not in the vasculitis group. In vasculitis patients, there was also no significant overall increase in neutralising antibodies against the WT, delta, or omicron BA.1 variant. Although 57.1% of patients produced SARS-CoV-2-specific antibodies, only half of them (28.6%) had detectable neutralising antibodies against WT and delta. No effective neutralising capacity of antibodies against omicron was found in the majority of cases. This is important because neutralising antibodies inversely correlate to the risk of symptomatic SARS-CoV2 infection [23], meaning that with regard to the currently prevalent omicron variant, patients with fewer neutralising antibodies are less protected from symptomatic courses. Vasculitis patients received immunosuppressive therapy, as shown in Table 1, which negatively affected the immune response to vaccination. Rituximab therapy leads to reduced antibody titers after SARS-CoV-2 mRNA vaccination due to B-cell depletion (at least months after the last administration) [24,25,26]. A significantly lower number of neutralising antibodies was also found during glucocorticoid and/or antimetabolite therapy compared to immunocompetent persons [12]. Carruthers et al. found seroconversion in 73.3% of vasculitis patients with renal involvement despite rituximab therapy [27]. By contrast, we could not detect antibodies in any patient who received rituximab over the previous 6 months (but also not in one patient on cortisone/mycophenolate therapy). This may be due to B-cell depletion, which correlates with the antibody count [28]. In contrast to our study, neutralising antibodies against different virus variants and T-cell immunity were not investigated by Carruthers et al. [27]. Vaccination-induced T-cell responses were observed (despite rituximab therapy), where IFNγ production was lower upon additional immunosuppressive therapy with prednisone [26]. It was reported that patients with (eosinophilic) granulomatosis with polyangiitis, despite B-cell depletion and lack of humoral response, had a positive T-cell response to S peptides in 90.9% of cases, which was almost comparable to immune responses in healthy individuals [29]. In another study, the T-cell response was reduced in 58% of patients with rituximab therapy and remained stable for up to five weeks after the second vaccination [28]. Notably, our results showed that the T-cell response was positive for S1 peptide in 71.4% of patients. Importantly, we observed strong background T-cell activity in those subjects with autoimmune disease not linked to a specific response to SARS-CoV-2 S proteins, which limits the interpretation of such T-cell assays regarding the presence of a vaccine-specific response. While a certain percentage of patients with CKD showed a specific T-cell response to SARS-CoV-2 antigens, this was not found in subjects under immunosuppressive therapy.

In comparison to other studies, our results are derived from longitudinal observation with a follow-up at 3 months after the first vaccination (mean of two months after the second vaccine administration). This study design allowed us to make a direct comparison with the baseline, in contrast to the studies mentioned so far. The time to follow-up, however, could lead to divergent results. Carr et al. assumed a peak of antibody titres in renal pre-dialysis patients at approx. 4 weeks after vaccination and assumed the greatest significance when measured >4 weeks after the second dose, which is also the case in our study [30]. Comorbidities such as cardiovascular, cerebrovascular, and primary renal disease were not associated with antibody response, as recently shown by Buchwinkler et al. [9].

To generate a better immune reaction and counter the effect of immunosuppressive drugs that inhibit protection from SARS-CoV2 infections, the American College of Rheumatology advises vaccinating 4 weeks before the next rituximab application if the risk of COVID-19 is low for the patient [31]. Our results indicate that an adaptation of the vaccine to current virus variants is necessary to optimise the immune reaction and provide sufficient protection from neutralising antibodies. 

The T-cell response increased a few days after the first vaccination and reached a peak 5–7 days after the second vaccination, with a decrease almost to values before the second vaccination after 80–120 days [32,33]. Thus, a more sustained T-cell response could have been missed by us in patients under immunosuppression. Our vasculitis cohort is heterogeneous (6 of 7 patients ANCA-positive, different immunosuppressive therapies) which may influence the results among this small number of participants. The T-cell reactions already before vaccination could be explained—at least partly—by cross-reactivity with other coronavirus strains, as already described in several studies [17,34].

Despite severe immunosuppression (10 patients on triple therapy and 7 on dual therapy), a significant immune response was detectable in KT patients. Neutralising antibodies increased significantly against the WT and delta variants, but seven KT patients showed no response (41.2%). Rincon-Arevalo et al. suspected that the immunosuppressive therapy on T-helper cells disturbed B and plasma cell differentiation [35]. Tacrolimus is known to inhibit signal transduction and cytokine release in T-cells, mycophenolate is associated with reduced seroconversion, whereas cortisone does not seem to have a major effect on seropositivity [36]. Previous studies reported a response to mRNA vaccination in 3 to 69% of KT patients [30,37]. Our results are significantly higher (76.5%) with a lower number of neutralising antibodies (52.9% for the WT). Immunosuppressive effects are dependent on several factors, such as age, number of immunosuppressive drugs, type of drug, time interval to transplantation, and BMI [37,38,39]. In contrast to other studies, the measurement after the second dose was later, and the patients were younger. Further, our analysis evaluates only the mRNA-1273 vaccine, which leads to higher rates of seroconversion than the BNT162b2 (Pfizer) mRNA vaccine [38]. Depending on the regimen used for immunosuppression, the response may differ in general but also with regard to the relative effects towards B- and T-cell responses. 

CKD patients produced significantly more antibodies than vasculitis and KT patients. Similar results were observed in a few studies comparing only CKD and KT patients [9,36], although in our study CKD patients did not require dialysis. It is known from other vaccines that both the cellular and humoral immune response is lower in patients with CKD and decreases with the severity of renal impairment [36,40,41]. In contrast to the vasculitis and KT patients, none of the CKD patients in our study were receiving immunosuppressive drugs. In addition, we detected neutralising antibodies against the WT and delta variants in 100% of CKD patients, whereas only 38.9% had neutralising activity against omicron. In addition, we observed a significant increase in T-cell response after the second vaccination in CKD patients, indicating a good response of the humoral and cellular immune system to SARS-CoV-2 vaccination. Unfortunately, there is no comparison with the healthy general population in our study, and the threshold values for sufficient immune protection are not defined and largely differ depending on the methods and tests used. 

## 5. Conclusions

In conclusion, the mRNA-1273 vaccine generates a significant humoral and cellular immune response after two doses in CKD patients not requiring dialysis. This is in sharp contrast to the results obtained in vasculitis and KT patients. KT patients show a significantly weaker immune response than CKD patients in terms of antibody concentrations and neutralising activity, which is almost absent against the omicron variant. Importantly, patients with vasculitis and after KT have hardly any detectable T-cell response to SARS-CoV-2-specific antigens although tests are often positive, which can be traced back to a high, unspecific background activity of T-cell activation. Vasculitis and KT patients are poorly protected from infection and severe disease, putting them at the front of the line for receiving immediate anti-viral treatment in case of infection [42].

## Figures and Tables

**Figure 1 microorganisms-11-01756-f001:**
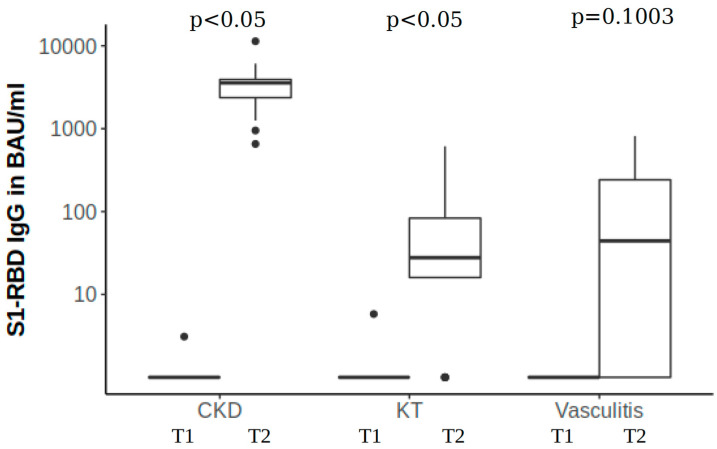
S1-RBD antibody response. Shown are the median of antibody titers at baseline (T1) and 3-month follow-up (T2) with 25% and 75% percentiles. *p*-values compare the number of antibodies in each disease group.

**Figure 2 microorganisms-11-01756-f002:**
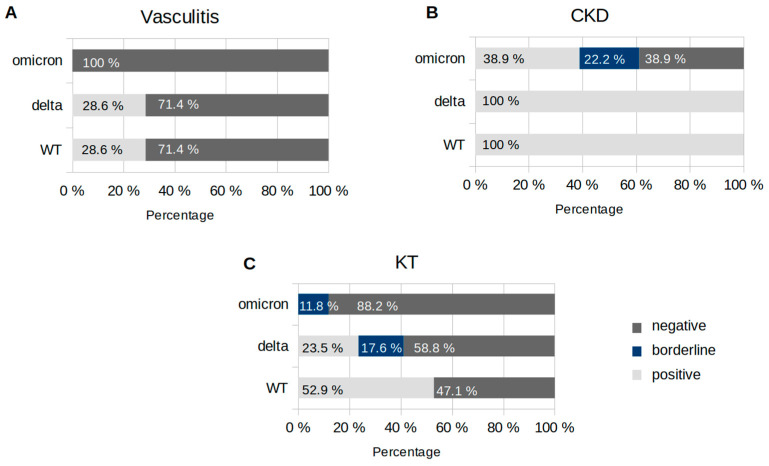
Relative number of patients with positive, borderline, and negative 50% neutralising antibody titres (NT50) three months after first vaccination with mRNA-1273: NT50 values are presented considering the respective underlying disease ((**A**–**C**) vasculitis, CKD, and KT) and virus variant (WT, delta, and omicron). NT50 values of ≥1:32 were interpreted as positive, 1:32 > NT50 ≥ 1:16 as borderline, and NT50 < 1:16 as negative.

**Figure 3 microorganisms-11-01756-f003:**
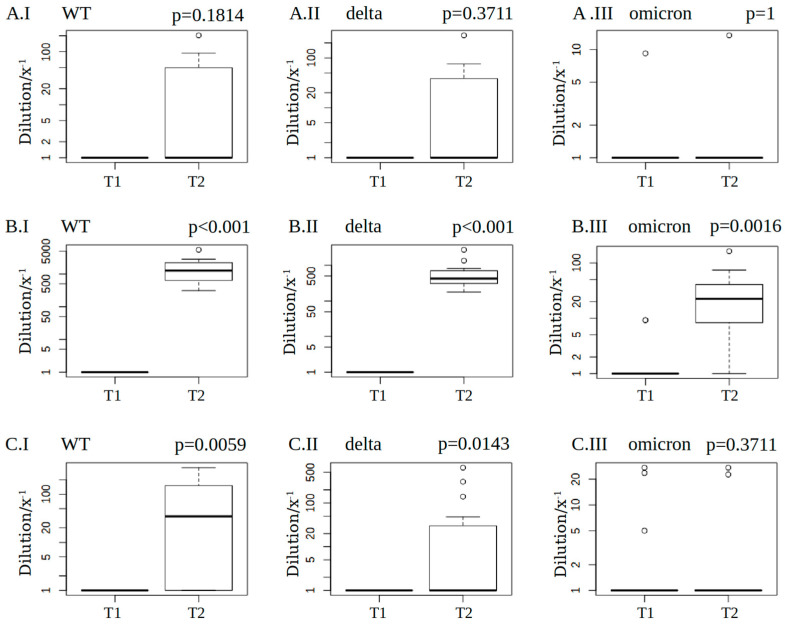
NT50 neutralisation values of patients vaccinated with mRNA-1273: The median NT50 values plus 25% and 75% percentiles at T1 (the day of the first vaccine dose) and T2 (three months later) are presented considering the underlying disease (**A.I**–**A.III** = vasculitis, **B.I**–**B.III** = CKD, and **C.I**–**C.III** = KT) and virus variant (I = WT, II = delta, and III = omicron). Minimum and maximum whiskers are shown, and outliers are identified if greater/less than the third quartile ±1.5 interquartile range. Wilcoxon signed-rank test for related samples was used to calculate *p*-values.

**Figure 4 microorganisms-11-01756-f004:**
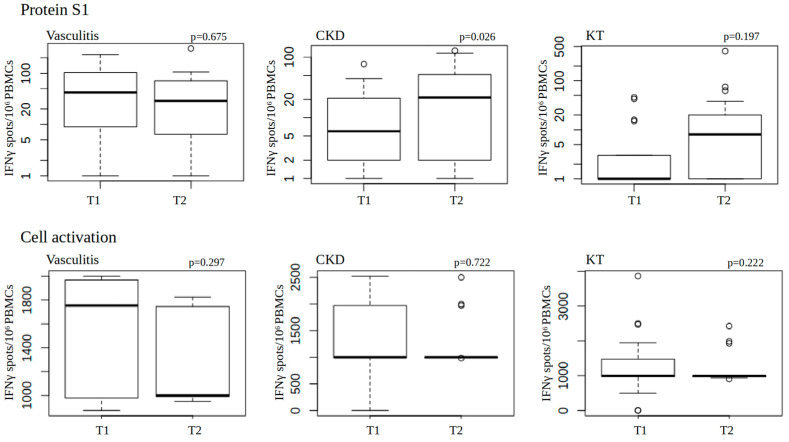
Specific T-cell response to protein S1 antigen and cell activation cocktail (1:500): mean cell activation on the day of vaccination and at follow-up. The *p*-value shows the comparison between the values at both time points in the Wilcoxon signed-rank test at T1 (on the day of the first vaccine dose) and T2 (3 months later and 2 months after the administration of the second vaccine dose) considering the disease. Boxplots represent the median, 25–75% quartiles, as well as minimum and maximum whiskers. Outliers are identified if greater/less than the third quartile ±1.5 interquartile range.

**Table 1 microorganisms-11-01756-t001:** Patients’ characteristics.

	CKD	Vasculitis	KT	Total
Total number of patients	18	7	17	42
Age of patients *	54 (11)	66 (11)	49 (12)	54 (13)
Number of men/women	9/9	4/3	9/8	22/20
Time interval between vaccine doses 1 and 2 * in days	43 [29; 44]	30 [29; 30]	30 [29; 44]	30 [29; 44]
Time interval between 1st vaccination and follow-up * in days	91 [90; 91]	91 [90; 91]	91 [90; 91]	91 [90; 91]
Spike antibodies before vaccination ** BAU/mL	0	0	0	0
Spike antibodies at follow-up *** BAU/mL	3572.9 [2376.8; 3930.1]	44.3 [0.0; 256.5]	27.8 [16.0; 83.3]	487.6 [27.1; 3147.6]
Number of patients under immunosuppressive therapy Mycophenolate Tacrolimus Azathioprine Cyclosporine Cortisone Rituximab	0	6 1 0 2 0 2 2	17 15 14 1 1 13 0	23 16 14 3 1 15 2
eGFR [mL/min/1.73 m^2^] ***	39 [31; 42]	51 [50; 61]	49 [43; 61]	44 [36; 57]

* Data are shown in days as mean (SD) or median [min; max]; ** the number of patients with positive spike antibodies (quantitative antibodies are considered positive from a threshold of ≥7.1 BAU/mL); *** data are shown as median [25% percentile; 75% percentile].

**Table 2 microorganisms-11-01756-t002:** T-cell response to SARS-CoV-2 S1 antigen on the day of the first vaccine dose (T1) and at three-month follow-up (T2).

Disease Cohort	T1	T2
Vasculitis	42 ± [9; 103,5] (71.4%)	29 ± [7; 71] (71.4%)
CKD	6 ± [2; 20] (38.9%)	22 ± [4; 49] (72.2%)
KT	1 ± [1; 3] (23.5%)	8 ± [1; 20] (47.1%)

Data are shown as median ± [25% percentile; 75% percentile] (percentage of positive T-cell response).

## Data Availability

Data are partialy available in Appendix A.

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
