# Peer review of "Virus-Subtype-Specific Cellular and Humoral Immune Response to a COVID-19 mRNA Vaccine in Chronic Kidney Disease Patients and Renal Transplant Recipients"

_microorganisms, 2023, doi:10.3390/microorganisms11071756_

Round 1

Reviewer 1 Report

The purpose of this study was to investigate the cellular and humoral immune response to mRNA-1273 vaccination in patients with vasculitis, chronic kidney disease (CKD), and kidney transplant (KT) recipients over a longitudinal period. The authors evaluated the immune response before vaccination and two months post-boost. The data showed that compared to vasculitis and KT patients, CKD patients had higher levels of antibodies and specific T cell responses induced by vaccination. Additionally, the antibodies in CKD patients had stronger and wider neutralizing activity against different variants of SARS-CoV-2.

The study's results provide valuable information for understanding the relationship between the humoral and cellular immune response and the host immune status, which can aid in preventing and controlling virus-related diseases. However, the manuscript would benefit from improved figure plotting and more thorough data analysis to better present and interpret the findings.

One significant limitation of this study is the lack of a healthy control group for comparison. Without proper controls, it is challenging to draw accurate conclusions about the specific role of immune response in CKD patients. Additionally, the authors did not analyze the immune status of patients receiving immunosuppressive therapy before vaccination, which would provide more meaningful conclusions from the study.

To improve the manuscript, the authors should address the following concerns:

1. Plot the Spike antibody titers before vaccination and at follow-up to provide a more detailed analysis of the immune response.

2. Revise Figure 2 and 3 with a clear y-axis title and revise the y-axis to log scale, label the statistical test results in the figures to make them more understandable.

3. Explain the reason for separating the data of vasculitis and CKD patients from KT recipients in Table 2.

Author Response

We thank the reviewer for the positive comments on our manuscript and the insightful comments:

1. Plot the Spike antibody titers before vaccination and at follow-up to provide a more detailed analysis of the immune response.

We have added a new Figure 1 reflecting titer concentrations.

2. Revise Figure 2 and 3 a clear y-axis and revise the y-axis to log scale, label the statistical results in the figures to make them more understandable.

According to the suggestions of the reviewer the y-axis was revised and transformed into a log scale. The p-values pre- and post vaccination are shown at the top of each figure.

3. Explain the reason for separating the data of vasculitis and CKD patients from KT recipients in Table 2

A graphical error occurred while converting the file for the journal, it has been corrected.

Reviewer 2 Report

 Line 170 "in vitro"

Adjust the resolution of Figure 1

Table 2 “Vasculitis”

Adjust the legends of the ordinate axis in the graphs of Figures 2 and 3

In Figure 3, check the legends of the axis of the abscissas.

Discuss further how low omicron neutralization by the serum of vaccinated patients could affect their protection and how it could be resolved. Remember that this strain and its variants are the most prevalent in the world today.

None

Author Response

We thank the reviewer for the positive comments on our manuscript and the insightful comments:

Line 170 "in vitro"

We added “in vitro” in the text.

Adjust the resolution of Figure 1

Resolution of all Figures adjusted

Table 2 “Vasculitis”

changed

Adjust the legends of the ordinate axis in the graphs of Figures 2 and 3

Ordinate axis of both graphs were integrated into the figures.

In Figure 3, check the legends of the axis of the abscissas.

Missing “T’s” were inserted into the Firgure at the abscissas.

Discuss further how low omicron neutralization by the serum of vaccinated patients could affect their protection and how it could be resolved. Remember that this strain and its variants are the most prevalent in the world today.

We have added a short paragraph in the discussion section (line 326-29 and 371-77 see section „discussion“)

Reviewer 3 Report

I think the topic is intersting but not so relevant and the paper must be revised. CKD is hetrogenic population? which stage? what eGFR range? mean GFR? primaryy renal disease ? did you observe any differences among different CKD stages? were they represented? did CKD patients had additional ciomorbidities that could contribute to immune response and disease severity?

many studies evaluated humoral response among CKD and dialysis population and should be cited

AAV - in which phase? induction? maintanance? 

did you assesed CD20? 

when you categorized by disease and not by treatment it could lead to bias and mis- interpatation of the results 

many studies evaluated humoral response among KT patients - should be cited and descibed as well 

abstarct should be revised 

methods section is not detalied and categorized 

did patients signed informed consent? 

since the above population were vaccinated 3-4 times I think you must highlight the novelty and rellevant of your finding in the discussion section 

moderate changes 

Author Response

We thank the reviewer for the insightful comments on our manuscript:

CKD is hetrogenic population? which stage? what eGFR range? mean GFR? primaryy renal disease? did you observe any differences among different CKD stages? were they represented? did CKD patients had additional comorbidities that could contribute to immune response and disease severity?

GFR is shown in the article as median + percentiles (see Table 1) as well as median CDK stage (see “3. Results"). We added absolute numbers of patients for each CKD stage (see “3. Results” line 196). However, most patients were in stage 3, while only a minority were in stage 4 (n=3)  or stage 2 (n=1) which precluded a statistical comparison of immun responses among CKD patients.  As Buchwinkler et al. 2021 showed comorbidities as cardiovascular and cerebrovascular disease as well as primary renal disease were not associated with antibody response. A short paragraph on that issue is added in section „Discussion“ (line 368-70).

many studies evaluated humoral response among CKD and dialysis population and should be cited

There are several studies which investigated the humoral immune response in CKD patients and we have cited a few important of them (Krueger et al. Am J Kidney Dis 2020; Ma et al. Front Med (Lausanne) 2022; Buchwinkler et al. J Clin Med 2021), but we were not able to include all of them due to restrictions of space.

AAV - in which phase? induction? Maintanance?

Phase of AAV patients: all were in maintenance state (information is added in "3. Results“ line 197).

did you assesed CD20?

There were only two patients under anti-CD20 therapy. B-cell numbers were not accessed in those patients.

when you categorized by disease and not by treatment it could lead to bias and mis-interpatation of the results

We have indicated the current immuno-suppressive therapy of patients in the respective groups in Table 1. This is a relevant point to be studied in future trials or in meta-analyses to access the effects of the respective immune-suppressive drugs on the outcome in specific diseases. Of note, a recent study compared the effects of different immunosuppressive drugs along different rheumatological diseases on vaccine response and specifically boosters (Bjorlykke et al. Lancet Rheumatol 2022) which gave hints on the effects of specific drugs

many studies evaluated humoral response among KT patients - should be cited and descibed as well

We added some references (Crespo et al. Am J Transplant 2022; Stumpf et al. Lancet Reg Health Eur 2021) but there are several studies which investigated the humoral immune response in KT patients and we already cited a important review (Carr et al. Kidney Int Rep 2021), but we were not able to include all of them due to restrictions of space.

abstarct should be revised

We gave some additional informations in the abstract.

methods section is not detalied and categorized

We added sections and gave further informations (see methods)

did patients signed informed consent?

Already in the article, see section 2.1 (line 101).

Since the above population were vaccinated 3-4 times I think you must highlight the novelty and releevant of your findings in the discussion

The novelty is the parallel assessment of humoral and cellular immune response and their relevance to neutralisiation of different SARS-CoV2 variants. This has been explained in the text and discussion.

Round 2

Reviewer 3 Report

the authors did not address my concerns

OK 

Author Response

We have responded to all the questions and suggestions given by the reviewer in the first round.   There is no detailed or specific information on what the reviewer further requests.